# Gene Regulatory Network Analysis of Perivascular Adipose Tissue of Abdominal Aortic Aneurysm Identifies Master Regulators of Key Pathogenetic Pathways

**DOI:** 10.3390/biomedicines8080288

**Published:** 2020-08-14

**Authors:** Luca Piacentini, Mattia Chiesa, Gualtiero Ivanoe Colombo

**Affiliations:** Immunology and Functional Genomics Unit, Centro Cardiologico Monzino, IRCCS, 20138 Milan, Italy; mattia.chiesa@cardiologicomonzino.it (M.C.); gualtiero.colombo@cardiologicomonzino.it (G.I.C.)

**Keywords:** gene regulatory network, transcription factors, perivascular adipose tissue, immune response, inflammation, abdominal aortic aneurysm, vascular diseases

## Abstract

The lack of medical therapy to treat abdominal aortic aneurysm (AAA) stems from our inadequate understanding of the mechanisms underlying AAA pathogenesis. To date, the only available treatment option relies on surgical intervention, which aims to prevent AAA rupture. Identifying specific regulators of pivotal pathogenetic mechanisms would allow the development of novel treatments. With this work, we sought to identify regulatory factors associated with co-expressed genes characterizing the diseased perivascular adipose tissue (PVAT) of AAA patients, which is crucially involved in AAA pathogenesis. We applied a reverse engineering approach to identify cis-regulatory elements of diseased PVAT genes, the associated transcription factors, and upstream regulators. Finally, by analyzing the topological properties of the reconstructed regulatory disease network, we prioritized putative targets for AAA interference treatment options. Overall, we identified NFKB1, SPIB, and TBP as the most relevant transcription factors, as well as MAPK1 and GSKB3 protein kinases and RXRA nuclear receptor as key upstream regulators. We showed that these factors could regulate different co-expressed gene subsets in AAA PVAT, specifically associated with both innate and antigen-driven immune response pathways. Inhibition of these factors may represent a novel option for the development of efficient immunomodulatory strategies to treat AAA.

## 1. Introduction

Despite intensive efforts over the last decades, abdominal aortic aneurysm (AAA) remains an elusive disease for which no effective treatment aiming to hinder or reduce its growth is yet available [1,2]. This reflects our incomplete understanding of the etiology and pathogenetic mechanisms leading to the development and evolution of AAA. Mechanistic studies mainly rely on animal models, which despite the large number of different methods available to induce an “acute” form of AAA, have not been individually able to thoroughly elucidate the pathogenesis of the disease, which instead presents a natural history characterized by a multifactorial, slow, and chronic process [3]. Thus, integration of in vivo models with knowledge from preclinical research on human patients is needed for a greater understanding of the processes underlying AAA [2].

A successful approach to dissecting complex phenotypes is molecular profiling, which allows a large-scale exploration of pathological processes in diseased tissues without the need for an a priori selection of the factors to be tested; therefore, it is potentially capable of uncovering new and unrecognized causes of disease onset or progression [4,5]. We have recently used this approach to explore the transcriptome of the perivascular adipose tissue (PVAT) in patients with abdominal aortic diseases [6,7]. PVAT has received growing interest in the study of large artery diseases, either because it has a fundamental role in the regulation of vascular physiology [8,9,10] or because its dysfunction is recognized to affect the development of both dilated and atherosclerotic aortic diseases [11]. Indeed, by comparing the adipose layer of a dilated abdominal aorta with that of a non-dilated aortic neck in each patient, we revealed locally restricted gene expression patterns characterizing the dilated PVAT of AAA. Overall, these genes were functionally associated with inflammatory and innate or adaptive immune responses, which along with other relevant pathways, including cell-death and extracellular matrix degradation, led us to hypothesize that AAA is an immunological disease with a possible underlying autoimmune component [6]. However, the regulatory elements underlying the pivotal pathogenetic processes associated with PVAT in AAA patients remained to be defined.

Reverse engineering from RNA expression data is a valuable and grounded approach that allows reconstruction of gene regulatory networks by identifying cis-regulatory elements, which are the targets of sequence-specific transcription factors (TFs) [12,13]. TFs are often defined as the master regulators of cellular processes because they can control the simultaneous expression of many genes at once [14]. TFs regulate the transcription of their target genes by interacting with other TFs and co-factors to constitute transcriptional complexes, but can also be directly targeted by other regulators acting upstream of the TFs, e.g., protein kinases. Identifying cis-regulatory elements and their respective TFs from co-expressed genes can, thus, provide important insights into the regulatory mechanism affecting a specific biological process.

In the present work, we aimed to identify the regulators that may control the most prominent pathogenetic processes we found associated with PVAT in AAA patients. For this purpose, we first applied a reverse engineering approach on co-expressed genes, distinguishing the dilated PVAT (diseased) from the non-dilated (healthy) aortic neck to find cis-regulatory elements and associated TFs. Then, we sought upstream regulators that could directly affect (e.g., activate) the TFs identified above or help form transcriptional complexes. Finally, exploring the topological properties of the resulting regulatory network, we outlined the most likely putative targets for AAA interference treatment options.

## 2. Materials and Methods

### 2.1. Study Population and Gene Expression Data

This study relies on the cohort of 30 AAA patients characterized in the previous study by Piacentini et al. [6]. Patient features, including demographic data, risk factors, medications, and exclusion criteria, have been extensively described in the aforementioned work. AAA patients enrolled in the study underwent elective surgery at the Centro Cardiologico Monzino IRCCS, Milan, Italy, between 2010 and 2014. Elective repair of AAA was done in compliance with the international and national guidelines for the care and treatment of AAA [15,16]. From each patient, adipose tissue specimens were collected at the time of surgery, which included the periaortic adipose tissue obtained from the aortic neck proximal to the aneurismal sac (non-dilated PVAT) and periaortic adipose tissue surrounding the aneurysmal sac (dilated PVAT).

The expression data produced in the study have been made publicly available at the NCBI’s GEO repository and can be accessed at https://www.ncbi.nlm.nih.gov/geo/query/acc.cgi?acc=GSE119717.

### 2.2. Selection of Differentially Expressed (DE) Genes in Dilated PVAT of AAA

Based on our previous results, we selected 172 unique genes positively associated with dilated AAA, corresponding to the most significant set of DE probes (transcripts), which distinguished dilated (diseased) from non-dilated (healthy) PVAT. Appendix A shows the annotation of these 172 DE genes. We used these DE genes as the seed gene list for the identification of cis-regulatory elements and candidate TFs.

### 2.3. Identification of Cis-Regulatory Elements and TFs

To identify regulatory elements, we relied on a reverse engineering approach to infer transcriptional regulatory network underlying the 172 DE genes by cis-regulatory sequence analysis. This analysis was performed in the Cytoscape environment v3.7.1 [17] through the iRegulon software v1.3 [18]. Briefly, iRegulon performs a “rank-and-recovery” procedure. The ranking step allows genes (RefSeq annotation) to be ranked for a library of positional weight matrices (PWMs), which represent matrices of regulatory motifs. Then, for each gene a regulatory search space around the transcription start site (TSS) is scanned for homotypic cis-regulatory modules (CRM) using a hidden Markov model across multiple vertebrate species. A ranked list of genes is generated, with the most likely genomic target of a specific motif at the top of the ranking. In the recovery step, the enrichment of genes from the input gene list (i.e., DE genes) is tested against the gene rankings by calculating the area under the cumulative recovery curve (AUC) in the top of the ranking (3%), which is then normalized into a normalized enrichment score (NES). The method exploits a wide collection of 18 databases of 9713 non-redundant TF motifs and 3 databases of 1120 chromatin immunoprecipitation sequencing (ChIP-seq) signals along the genome (i.e., tracks).

Parameters used for the ranking step were: motive collection: “10K (9713 PWMs)” (i.e., the most extensive motif collections); track collection: “1120 ChIP-seq tracks (ENCODE raw signals)” (i.e., full collection of ChIP-seq data against TFs); putative regulatory region: “20 kb centered around TSS” (i.e., it may return promoter-based and/or distal regulators); motif rankings database: “20 kb centered around TSS (7 species)” (including the mammalian species *Bos taurus, Canis familiaris, Mus musculus, Monodelphis domestica, Pan troglodytes, Macaca mulatta*, and *Rattus norvegicus*, as well as considering the conservation among them).

Parameters for the recovery step were: enrichment score threshold = “3.0” (raising the post-hoc threshold to 3.5); receiver operating characteristic (ROC) threshold for AUC calculation = “0.03”; rank threshold = “5000”; minimum identity between orthologous genes = “0”; maximum false discovery rate (FDR) on motif similarity = “0.001”.

Resulting motifs and tracks were ranked according to NES and labeled with the motif or track identifier (ID) of their original database. Motifs that shared a higher level of similarity were grouped into clusters using the default method (see [19]). The motif-to-TF association procedure for the algorithm, based on motif similarity and orthology, returns the TFs that more likely bind to the enriched motif. The TFs with the highest levels of confidence (i.e., TFs recorded as having a direct annotation, meaning that the PWM was determined for a certain TF in that species) were selected as the most reliable candidate TFs for each cluster of motifs or tracks.

### 2.4. Inferring the TFs’ Upstream Regulatory Factors

To infer regulatory modules upstream of the “hub” candidate TFs, additional proteins that directly connect to TFs were firstly identified (i) by using experimentally reported protein–protein interactions (PPI) or protein complexes; and then (ii) by identifying protein kinases regulating the above extended transcriptional complexes. The Expression2kinase (X2K) software v1.6.1207 [20] was used with default parameters for both protein network expansion and kinase retrieval, except for the “allow a maximum of 10,000 node links from a node”, “allow a maximum of 100,000 interactions from an article”, and “allow a minimum of 1 article reporting a specific interaction” options, which were enabled to ensure a higher quality of interactions in outputted additional proteins. The method for drawing PPI exploits experimentally validated mammalian interactions from 18 databases containing more than 24,000 proteins and almost 390,000 interactions. The kinase–substrate interactions are instead from other sources for a consolidated dataset of 14,374 interactions from more than 3400 publications on 436 kinases (see [20] for details).

### 2.5. Topological Analysis

Topological analysis was performed on two different networks through Network Analyzer v3.3.2 [21] and CentiScape 2.2 software [22] into the Cytoscape environment v3.7.1. The first analyzed network was the whole regulatory network (cf. Figure 2) drawn to connect candidate TFs (regulators; source nodes) with their target DE genes (regulated; target nodes). This facilitated visual inspection of clusters of shared and unique target regulated genes and their relationships. To disentangle the complexity of the network and to find the most relevant “hub” TFs, three centrality measures, i.e., connectivity degree (aka degree), betweenness, and radiality, were evaluated. The higher the values of these topological indexes among nodes, the higher the relevance of a specific node in the network in relation to the others [23]. TFs ranked in the upper tertile for the three centrality measures were selected as the “hub” master regulators.

The second topological analysis was performed on the regulatory transcriptional complex network made of “hub” TFs, additional or intermediate proteins, and upstream kinases (cf. Figure 3). The network was treated as a directed network because the regulator–regulated target relationships were highlighted by directed edges. Measures of in-going and out-going direct connections (edges) of a node were defined as the in-degree and out-degree, respectively. Additionally, radiality, stress, betweenness, bridging, centroid, closeness, eccentricity, eigenvector centralities were also assessed. The relevance of the topological index for each node was weighted through the z-score calculation, which assesses how a specific observation (node) moves away from the mean of the total observations, measured in terms of standard deviations from the mean. A node with a z-score ≥2 was deemed as very relevant for that topological index.

For an easier interpretation of topological indexes [22,23], it should be considered that nodes with a large degree of connectivity are defined as “hubs”, since they are likely crucial factors that might play key (e.g., causative) roles in the biological context (e.g., disease) of interest [24,25]. Nodes with high betweenness (and with similar but not equal stress, bridging, closeness, and eigenvector values) are defined as “bottlenecks”, which are likely important factors that can hold together communicating proteins or genes and which might be relevant as organizing (or very central) regulatory molecules [26]. Nodes with high eccentricity show how easily a protein or gene can be functionally influenced by all other proteins or genes in the network. High centroid values instead show those nodes that are functionally able to organize subclusters of proteins or genes, thus possibly coordinating the activity of nodes with high connectivity. A node with high radiality, high eccentricity, and high closeness provides a consistent indication that it plays a central position in the network.

### 2.6. Linking Transcriptional Complex Clusters with AAA Pathogenetic Biological Functions

Functional association of transcriptional complex clusters (each including a subset of the seed list of DE genes and additional or intermediate proteins and kinases) with AAA pathogenetic biological processes or pathways was performed by measuring the overlap of the transcriptional complex cluster gene sets with the results of the original gene set enrichment analysis (GSEA; see Supplementary Materials of [6]). The collection of transcriptional complex cluster gene sets was generated in the Gene Matrix Transposed file format (*.gmt) and imported into the Enrichment Map software v3.2.1 [27] to visualize and measure the overlap of these gene sets with the enrichment network drawn from GSEA, which displayed all of the significant Gene Ontology–biological process (GO-BP) orpathway gene sets that were suggested to be pathogenically associated with dilated PVAT in AAA. The consistency of the overlap was tested through a hypergeometric test. Associations were deemed to be significant for adjusted *p*-values < 0.01 (Benjamini–Hochberg method for multiple testing correction) and with a gene set overlap corresponding to ≥5% of the genes for each specific transcriptional complex cluster. To facilitate visualization of the most significant relationships of each transcriptional complex cluster with their associated GO-BP or pathways, an enrichment subnetwork was subsequently generated.

## 3. Results

For this work, we selected the 172 unique genes we found overexpressed in dilated PVAT compared with non-dilated PVAT in AAA patients (see Supplementary Materials of [6]). Gene annotation is reported in Appendix A. These genes were positively associated with the layer of adipose tissue surrounding the dilated (diseased) abdominal aorta and included several factors of the inflammatory and immune responses, which were claimed as being the most relevant for the pathogenesis of AAA [28,29]. We, thus, sought the regulatory sequence motifs (i.e., target gene nucleotide sequences used to control its expression) and tracks (i.e., ChIP-seq signals along the genome) for these 172 genes and reconstructed a gene regulatory network.

### 3.1. Identification of Motifs and Tracks of Genes Over-Expressed in Dilated PVAT of AAA Patients

By cis-regulatory sequence analysis, which leverages the use of combined multiple collections of motifs and tracks, we identified 30 motifs and 1 track with NES values >3.5 and AUC values >0.06. The selected NES and AUC thresholds were chosen to reduce the probability of recovering false-positive associations, which would, therefore, ensure more robust and accurate results [18]. Exploring the motif enrichment results, we observed that subgroups of the 30 motifs showed a high sequence overlap, which were, thus, grouped into 8 different clusters based on similarity (M1 to 8; Appendix A). Notably, each cluster included one TF with a direct annotation, which represents the highest level of confidence for a motif-to-TF association. Thus, we ranked the TF with the direct annotation as the most reliable candidate TF for each related motif or track cluster (Table 1, Figure 1, and Appendix A).

#### 3.1.1. Regulatory Network of TFs and Over-Expressed Genes in Dilated PVAT

To visually explore the relationships between candidate TFs and their relative target genes, we drew a gene regulatory network consisting of TFs as source nodes, presenting direct evidence for significant motifs with the highest NES values, as well as the DE genes as target nodes (Figure 2). The network shows the complex relationships between TFs and their targets, with TFs displaying both shared and unique target genes.

#### 3.1.2. Selection of “hub” TFs through Topological Analysis

Although all of the identified TFs presented a significant association with DE genes, we tried to figure out the regulatory elements that more likely associate with the most relevant pathogenetic processes in dilated PVAT of AAA. For this purpose, we applied a network topology analysis to identify TFs representing the most relevant “hub” in the whole regulatory network. With this approach, we found that three TFs, i.e., TBP, NFKB1, and SPIB, displayed the highest values for all the assessed topological centrality measures (Table 2). The advantage of this approach is that we could extract key TFs by taking into account both the statistics of motif and track identification and TF “centrality” ranking in a complex gene regulatory network. In particular, the degree index indicates the relevance that SPIB, NFKB1, and TBP have within the network—they displayed the highest connectivity by interacting with large numbers of the 172 DE genes (104, 91, and 74, respectively), suggesting a central regulatory role.

### 3.2. Identification of Transcriptional Complexes and Upstream Regulators

We refined our search for regulatory molecules by including those that physically connect to and act on the identified “hub” TFs. To this end, we first extended the regulatory network by retrieving intermediate direct interactors of SPIB, NFKB1, and TBP, and by drawing putative transcriptional complexes. Then, we inferred upstream protein kinases that could regulate those transcriptional complexes.

#### 3.2.1. Connecting Additional Proteins to TFs through Protein–Protein Interactions (PPI)

By leveraging experimentally validated PPI, we identified 28 intermediate proteins (annotated in Appendix A) directly linked to SPIB, NFKB1, and TBP. Then, we drew a directional network to show the relationships (i.e., from regulator-to-regulated target) among the “hub” TFs and all the retrieved intermediate proteins (Figure 3). NKFB1 and TBP displayed the highest direct connections, with 8 out-degree (outcoming connections) and 19 in-degree (incoming connections) and with 12 out-degree and 15 in-degree connections, respectively. SPIB presented 4 out- and 3 in-degree connections but displayed a direct edge towards TBP, suggesting that their possible interaction constitutes a transcriptional complex together with other proteins. Notably, we found 18 other TFs among the additional proteins (i.e., AR, BCL3, CEBPB, CREBBP, E2F1, ELF3, ESR1, FOS, JUN, KLF5, NCOA1, NCOA6, NR3C1, REL, RELA, RXRA, SP1, and SPI1), which may participate in forming such transcriptional complexes (Figure 3; green ovals). Consistently, some of them were also predicted to associate with motif clusters M2 (i.e., BCL3, E2F1, REL, RELA), M3 (i.e., SPI1), and M8 (i.e., CEBPB; cf. Appendix A), strengthening the idea that these TFs may cooperate with “hub” TFs to target common sequence motifs and coordinate transcription of DE genes. Furthermore, relying on the Transcription co-Factor DataBase (TcoF-DB) [30], we could annotate 7 proteins that act as transcriptional co-factors (i.e., that may regulate transcription by interacting with TFs), but in contrast to TFs these do not bind directly to regulatory DNA regions (Figure 3; orange ovals). Six of them (i.e., CTNNB1, HDAC1, HMGB1, RUVBL2, SIN3A, and TRIP4) have experimental evidence for both involvement in transcriptional regulation and for presence in the cell nucleus (classified as class high-confidence); RUVBL1 has evidence: inferred from electronic annotation” (classified as class 2) for involvement in transcriptional regulation.

#### 3.2.2. Identification of Protein Kinases Upstream Transcriptional Complexes.

The expansion of the network to protein–TF interactions allowed an increase of the number of possible regulatory molecules that can influence specific functions in the dilated PVAT of AAA. The search for further kinase–substrate interactions can add an upstream level of control of gene expression and helps to identify key target regulators. Indeed, by kinase enrichment analysis, we found 42 associations with protein kinases (nominal *p*-values < 0.01), of which 28 stood correction for multiple testing (adjusted *p*-values < 0.01; Appendix A). To focus on the most significant kinases, we selected those that were ranked in the upper tertile of the distribution according to the “combined score”, i.e., CHUK, CSNK2A1, CSNK2A2, GSK3B, MAPK1, MAPK14, MAPK3, MAPK8, PRKDC, and TAF1 (Figure 3, light blue ovals; annotation in Appendix A).

Network analysis showed that CHUK, CSNK2A1, GSK3B, and PRKDC were directly associated with NFKB1; TAF1 targeted TBP; and CSNK2A1, CSNK2A2, GSK3B, MAPK3, MAPK8, and MAPK14 interacted with SPIB. Specifically, GSK3B, MAPK1, and CSNK2A1 displayed the highest out-degree values (17, 15, and 13, respectively), which suggests that they can be functionally relevant for most of the protein–TF interactions in the regulatory network.

#### 3.2.3. Topological Analysis of Regulatory Transcriptional-Complex

Topological analysis of the regulatory transcriptional complex network (cf. Figure 3), inferred the relative importance that specific proteins and genes may have as regulators in the network (Appendix A). First, we observed that centrality measures with similar meanings were highly correlated, indicating their consistency (Figure 4). Indeed, out-degree, radiality, and closeness, which indicate the possibility that a node is functionally relevant for several others, showed high positive correlations, with Pearson’s coefficient r values ranging from 0.89 to 0.99 (*p*-values < 0.001). Similarly, betweenness, bridging, and stress indexes, whose values if elevated suggest that a gene or protein likely connects (i.e., holds together) pivotal regulatory molecules, were positively correlated, with r values ranging from 0.65 to 0.97 (*p*-values < 0.001).

Specifically, we showed that NFKB1 showed the highest degree (total connectivity), in-degree, and eccentricity, suggesting that it is likely affected by the activity of many other proteins. Interestingly, REL, which is part of the well-known NK-kB complex together with NFKB1, has high eccentricity as well, although with lower global connectivity. NFKB1 also presents high betweenness and stress, suggesting the capability to hold together communicating proteins and to play a role as an organizing regulatory molecule. Similarly, RELA, which is also a subunit of the NF-kB complex, has the highest betweenness, stress, and bridging centrality values, leading to the hypothesis that it can serve in an organizing module along with NFKB1, “bridging” another possible set of regulatory proteins. For example, RELA targets TBP, which also displays very high betweenness and stress indexes and may serve as a central regulatory protein. Another remarkable finding concerns the role of MAPK1. Indeed, we can postulate that MAPK1 could be functionally essential for several other proteins, as indicated by its eigenvector index, out-degree (*n* = 15), and high closeness and radiality values compared to the mean of the network, which overall suggests a prominent regulatory role for this kinase in the network. Similarly, GSK3B showed a high out-degree (*n* = 17), radiality, and closeness over the network mean. Together these results provide a consistent suggestion that both MAPK1 and GSK3B kinases might have central regulatory roles in the network. CREBBP, as with NFKB1, RELA, and TBP, showed a high-stress index, indicating that it may be functionally relevant in connecting other regulatory molecules. CREBBP presents a high total degree of connectivity (15 in-degree and 13 out-degree connections), with a direct connection towards NFKB1 and incoming connections from RELA and SPIB. TBP displayed high stress and betweenness, which underlie its important roles as a “hub” TF and an organizing regulatory protein. Finally, RXRA showed the most consistent bridging index together with RELA, indicating its possible role as a connector of other central regulatory proteins. Indeed, RXRA is directly linked to very central proteins, such as the TFs NFKB1, RELA, and TBP, and the kinases MAPK1 and GSK3B. In line with this, we observed that the connection between RELA and RXRA showed the highest “edge-betweenness” value (33; with network mean ± SD = 8 ± 5, over 311 total edges), which indicates that this connection separates highly interconnected subgraph clusters.

### 3.3. Association of Regulatory Subnetworks with AAA Pathogenetic Pathways

We eventually built the entire regulatory network obtained by reverse engineering, including DE genes, candidate TFs, intermediate protein interactors, and upstream kinase regulators. By selecting the first neighbors of each “hub” TF throughout the network, we generated three “transcriptional clusters” to infer specific functional association with AAA-related pathways. This approach allowed matching each transcriptional cluster with those GO-BP or pathways associated with dysfunctional PVAT that could have a pathogenetic role in the development or progression of AAA (see Supplementary Materials in [6]). The significance of the overlapping genes, which measure the consistency of the association between each transcriptional cluster and specific GO-BP pathways, was estimated through a hypergeometric test. We found 69, 54, and 32 GO-BP pathways with a significant number of genes overlapping those of the transcriptional clusters of NFKB1, SPIB, and TBP, respectively (adjusted *I*-values < 0.01 and overlap size threshold ≥ 5% for the genes of the transcriptional cluster; Appendix A). Interestingly, the 3 transcriptional clusters presented both shared unique associations with pathogenetic GO-BP pathways. To summarize results and reduce redundancy of the overlapping GO-BP pathways, we drew an enrichment network (Figure 5). We found that the NFKB1 transcriptional cluster was uniquely associated with “regulation of lymphocytes proliferation” (specifically T-cells), “regulation of protein secretion”, and “vasculature development”. The SPIB transcriptional cluster was univocally associated with “regulation of phagocytosis”, “granulocyte or neutrophil chemotaxis”, “humoral immune response”, and “Fc receptor-mediated signaling”. TBP displayed a distinctive association with “regulation of cytokine’s biosynthetic process”. NFKB1 and TBP transcriptional clusters shared GO-BP pathways involved in TLR signaling (in particular, TRAF6-mediated induction of NFkB and MAP kinases upon endosomal TLR activation) and in the regulation of the JAK/STAT cascade. NFKB1 and SPIB transcriptional clusters shared GO-BP pathways involved in the regulation of lymphocyte activation and differentiation and inflammatory response, as well as in leukocyte cell–cell adhesion. Finally, we found common GO-BP pathways between the three transcriptional clusters, e.g., “response to molecule of bacterial origin”, “leukocyte chemotaxis”, “regulation of ERK1 and ERK2 cascade”, and “IL-4 and IL-13 signaling”.

## 4. Discussion

The pathogenic mechanisms responsible for AAA formation and expansion are largely unknown. PVAT is a well-known vessel homeostasis regulator that can have a major role in vascular disease pathogenesis [11,31]. Mounting evidence suggests that dysfunctional PVAT may play a role in vascular diseases, including AAA development [32]. To date, large-scale gene expression profiling of AAA has been carried out only on full-thickness aortic walls, excluding the adipose layer of the vessel [33]. Through genome-wide expression studies, we have recently gained insights into the important role that PVAT can play in both atherosclerotic and non-atherosclerotic abdominal aortic diseases in humans [6,7]. In AAA patients, our data suggested that an altered immune response in PVAT, alongside other concurring mechanisms, is a crucial pathogenetic element that leads to the progression of AAA, and ultimately to its rupture, by amplifying inflammation and degenerative mechanisms (e.g., loss of aorta structural integrity) [6].

With this work, we inferred the regulatory molecules that govern the most prominent pathogenetic processes in PVAT of AAA patients. Our study consistently shows that subsets of co-expressed genes characterizing the diseased PVAT in AAA present common cis-regulatory elements, which are the targets of specific TFs.

The reconstruction of regulatory networks proves to be particularly enlightening, as the disease phenotype is only occasionally the result of a single gene or protein effector, but is more commonly the combination of multiple pathobiological effectors acting together in complex relationships [24]. Analyzing molecular networks through their topological properties can help to identify both crucial disease genes (hubs) and potential drug targets [24,34]. However, it is worth noting that pharmacological targets are not necessarily the network “hub nodes”, because these are often essential factors with a strong influence on the phenotype of the cell or organism. Hub genes or proteins should be targeted only in particular subsets of cells with a strong and direct pathogenetic role. Otherwise, interference with the activity of essential genes may influence physiological processes that are crucial to the normal functioning of the organism. On the contrary, proteins that act as connectors (or presenting lower degrees of connectivity) have a lower impact on the overall structure of the network, thus resulting in likely safer candidate disease targets [35,36].

We identified SPIB, NFKB1, and TBP (i.e., “hub” TFs) as the master regulators of the resulting gene regulatory network, as they have the largest connectivity with the co-expressed genes associated with diseased PVAT. Therefore, we hypothesized that these TFs could be responsible for regulating the most impactful pathogenetic mechanisms of dysfunctional PVAT in AAA patients. Additionally, we identified additional proteins that can directly interact with the “hub” TFs to form transcriptional complexes, as well as protein kinases that can regulate the state of activation of “hub” TFs and transcriptional complexes.

We showed that the transcriptional clusters of SPIB, NFKB1, and TBP have strong functional associations with mechanisms of the innate and adaptive immune response, which we assumed to be the main drivers of the (auto)immune mechanisms in AAA PVAT. Specifically, we associated the NFKB1 transcriptional cluster with the positive regulation of lymphocyte proliferation, and together with TBP with the expression of genes involved in the innate immune response, i.e., the toll-like receptor (TLR) signaling. This is consistent with the well-known role of NF-kB complex activation in guiding both lymphocyte function, including cell proliferation and differentiation [37], and TLR signaling [38], which ultimately ends in NF-kB-triggered downstream transcription of genes that allow for an effective response to primary stimuli, such as antigens (or self-antigens) and danger- or pathogen-associated molecular patterns [39]. Additionally, NFKB1, REL, and RELA may differently combine to form heterodimers and carry out both unique and overlapping roles in T-cell proliferation during different stages of the cell cycle [40]. The relevance of the NF-kB complex in AAA pathogenesis has already been shown in in vivo AAA-induced models. Activation of the NF-kB complex has been proposed as a key factor inducing macrophage infiltration and osteoclastogenic differentiation [41,42] and affecting the inflammatory response in other cell types (e.g., vascular smooth muscle cells and mesenchymal cells) [43,44], ultimately leading to aortic inflammation and vessel wall degeneration. Nonetheless, our data suggest that lymphocyte activation and proliferation mechanisms could be crucial for amplifying the local antigen-driven immune response in the PVAT of AAA patients. Interfering with NK-kB signaling [45] in immune cells may, thus, have a significant impact on the evolution of AAA. Although immune-suppressive treatment should be carefully evaluated [46], selectively inhibiting NF-kB signaling in activated and proliferating lymphocytes in dysfunctional AAA PVAT may be a targeted therapeutic intervention without compromising the healthy effect mediated by NF-kB in other immune cells.

Mechanisms of the innate and adaptive immune response can also be mediated by SPIB and its cognate SPI1, which we found to be directly connected to SPIB in the regulatory network. Both SPIB and SPI1 are required for all of the signaling pathways by which B-lymphocytes sense and respond to local environmental stimuli, including antigens and molecules acting through TLRs [47]. Accordingly, we found that the SPIB transcriptional cluster was associated with response to TLR engagement, humoral immunity, and lymphocyte activation in PVAT of AAA. Interestingly, we also found a significant association with Fcγ receptor signaling, which underlies a localized antigen-specific humoral immune response. This signaling mechanism can be activated by B cells or neutrophils in the presence of immunoglobulin–antigen aggregates [48,49], and has been involved in the pathogenesis of AAA [50]. The downregulation of Fc receptor actions has been proposed as a therapeutic approach for other inflammatory and autoimmune diseases [51]. Targeting Fc receptor signaling regulators by manipulating the SPIB/SPI1 complex could be tested to block or minimize the local self-sustaining chronic inflammation observed in AAA PVAT.

Evaluating topological indexes of the extended regulatory network, we found that the protein kinases MAPK1 and GSK3B and the nuclear receptor RXRA (a type of retinoid X receptor) appear to play key roles in AAA pathogenetic mechanisms, and thus may be intended for interference therapy, as they are “non-hub” connecting proteins that could regulate signaling through the “hub” TFs. Both MAPKs and GSK3s have been recognized as having considerable roles in the immune response. MAPKs are strongly involved in innate immunity, including signaling related to TLRs [52]. Additionally, the MAPK/ERK pathway is an important modulator of matrix metalloproteinases during AAA formation in in vivo models, and its inhibition could represent a possible therapeutic approach to prevent AAA formation [53]. Consistently, we found that NFKB1, SPIB, and TBP transcriptional clusters were associated with the regulation of ERK1 and ERK2 cascade in the PVAT, reinforcing the concept that specific protein kinases play central roles in the regulation of inflammation and immune response in AAA. A direct role of GSK3s in AAA pathogenesis has not been established, although GSK3s were shown to regulate the production of pro- and anti-inflammatory cytokines and to influence the proliferation, differentiation, and survival of T-cells. GSK3s mainly work through the regulation of critical transcription factors, including NF-kB, the inhibition of which has been proposed for the treatment of several pathological conditions with underlying altered immune responses in animal models [54]. Evidence concerning the roles of retinoid X receptors in AAA is also limited [55]. It has been proposed that RXR activation may have a beneficial role in AAA by inhibiting the angiotensin type 1 receptor in vascular smooth muscle cells, a factor which is known to affect AAA development in angiotensin II–induced AAA models [56]. However, RXRA is also known to be involved in the regulation of innate immunity [57], and thus its inhibition could suppress immune-mediated pathogenetic mechanisms in AAA PVAT.

Since SPIB, TBP, MAPK1, GSK3B, and RXRA have not yet been associated with AAA in humans, they may be considered as newly identified targets for disease treatment.

Other proteins or kinases with less connectivity in the regulatory network may be potential therapeutic targets for AAA [35]. An example is histone deacetylase 1 (HDCA1), which despite its low centrality in our regulatory network, directly interacts with both NFKB1 and TBP, and thus might be considered as a possible target for interference treatments. Consistently, the use of HDACs inhibitors has been demonstrated to be effective in in vivo AAA models and has been proposed for treatment in humans [58]. Notably, HDACs have many TFs as natural substrates, including NF-kB, and are well-known regulators of the T-cell immune response. HDAC inhibition enhances T-regulatory cell survival and immune-modulating functions, and affects the development of T-cell response, including T-cell proliferation in response to antigen stimulation [59].

In summary, innate and adaptive immune responses are characterized by complex crosstalking mechanisms [60] and are both involved in AAA pathogenesis [28]. Additionally, communication between perivascular adipocytes and immune cells may also play a role in these complex relationships. For example, adipocytes secrete soluble factors (e.g., adipokines, chemokine, or pro-inflammatory interleukins), which may trigger signaling pathways in target cells involving the activation of the NF-kB complex, among others. Immune cells, in turn, produce inflammatory molecules and recognize antigens presented by the class I or II human leukocyte antigens (HLAs) expressed on adipocytes [8,61,62].

Innovative strategies for treating AAA may include both immune modulation to stimulate local anti-inflammatory mechanisms and targeting of specific pathogenetic lymphocytes and their interactions with other cells, such as antigen-presenting cells [63,64,65]. In our previous research, while showing that the activation of several pathways can contribute to the evolution of the disease, we assumed that immune response processes were essential to sustain a chronic inflammatory cycle that characterizes AAA, since they were involved in both the early and later stages of AAA [6]. It is tempting to speculate that interventions on these specific pathogenetic pathways could be therapeutically beneficial, because they could limit the main harmful mechanism that appears to be necessary for all phases of AAA. Notably, an immune phenotyping analysis in human AAA samples recently found that T-lymphocytes are the primary cell leukocyte population in AAA, with the largest concentration in PVAT, and that these PVAT-associated T-lymphocytes correlated with the severity of the disease [66]. Assessing the presence of a locally restricted, antigen-driven clonal expansion in PVAT of AAA could, thus, be the goal of future studies, which will allow for the precise detection of pathogenic lymphocytes for direct interference treatment [45], possibly through the above-suggested disease gene targets.

To our knowledge, this is the first study that describes the regulatory elements that may contribute to controlling the major pathogenetic processes observed in PVAT of AAA patients. Our study takes advantage of a reverse engineering approach that emerged as a valuable tool for elucidating cellular function and dysregulation in pathological contexts, and which promises to increase our ability to identify potential therapeutic targets and disease biomarkers as well [13].

Our work also has limitations. It relies on inferential analysis and suggests putative sensitive AAA target genes, for which formal evidence should still be provided showing that they can be used for effective interference treatments. Furthermore, our work has focused on the potential regulators of the most significant overexpressed genes restricted to the diseased PVAT of our AAA patient cohort, without ruling out the possibility that other regulatory mechanisms may still be important in the development and progression of the disease. Finally, we did not evaluate any possible interactions with genetic variants known to be associated with AAA [67]. However, given their putative relevance in aneurysm diseases, it may be important in future research to explore the existence of specific polymorphisms that could affect gene expression and offer insight into the molecular pathogenesis of AAA.

## 5. Conclusions

After a long-lasting period of research, our understanding of AAA is still in its “adolescence”. Despite a clear knowledge of the pathological hallmarks, the enigmatic “trait” characterizing AAA is still unknown, and is a relentless obstacle to its overall comprehension and the potential for effective care.

The translational gap between preclinical disease models and successful clinical studies prompts the research on AAA to explore and find novel candidates for treatments. Due to its critical role in AAA pathogenesis, PVAT promises to be a reliable target for testing innovative treatment options.

With this work, we identified master regulators of prominent pathogenetic processes associated with PVAT of AAA patients, i.e., altered immune response, including antigen-specific lymphocytes activation or proliferation and TLR signaling. Through the reconstruction of a gene regulatory network and the associated upstream regulators, we also suggested novel possible targets that may be considered for locally restricted interference treatments of AAA.

## Figures and Tables

**Figure 1 biomedicines-08-00288-f001:**
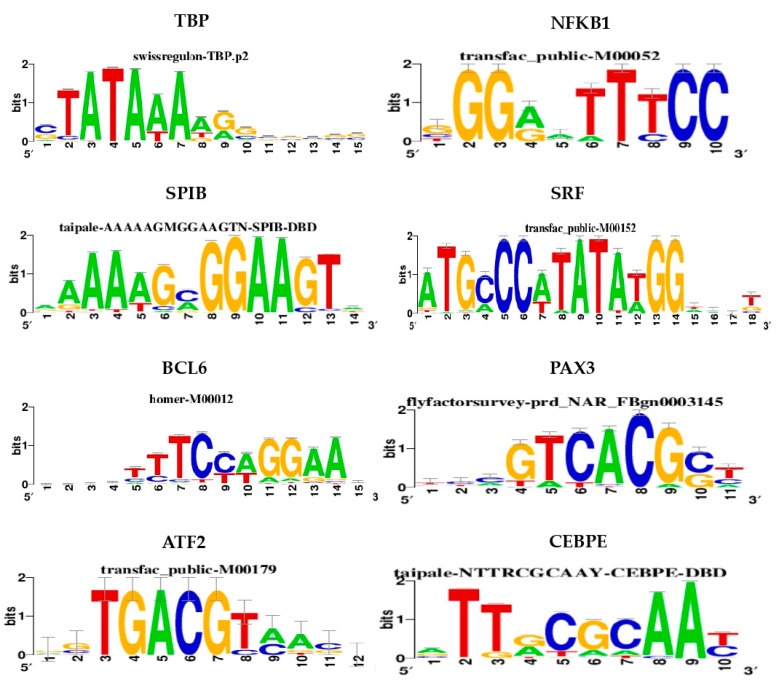
Target nucleotide sequences. Logos of cis-regulatory elements corresponding to the highest enriched motif of each cluster are shown. Motif IDs are reported on top of each sequence. The logo for tracks, i.e., CHD1, is not available.

**Figure 2 biomedicines-08-00288-f002:**
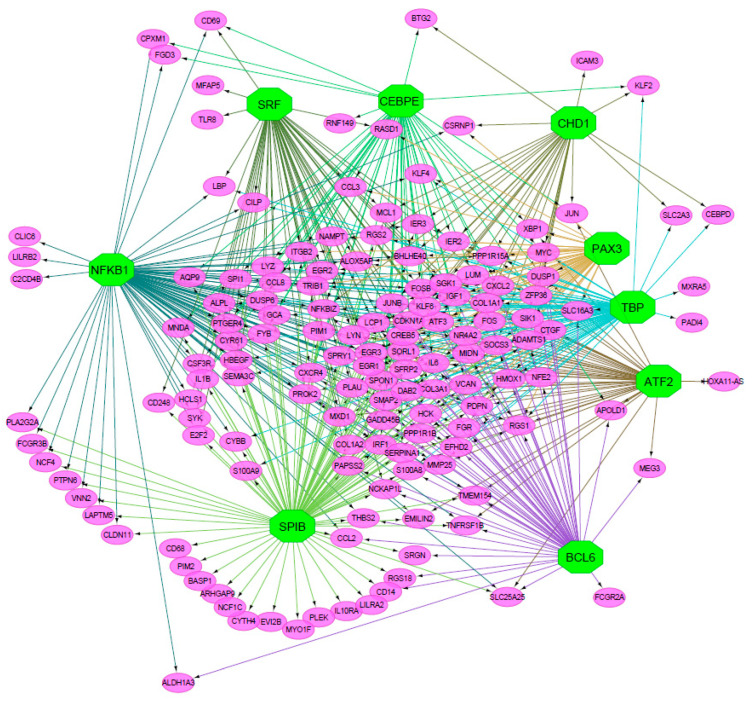
Gene regulatory network. The network shows the connectivity between candidate transcription factors (source nodes; green hexagons) and differentially expressed (DE) genes (target nodes; pink ovals). Edge color refers to clusters of DE genes based on the connections to a specific TF.

**Figure 3 biomedicines-08-00288-f003:**
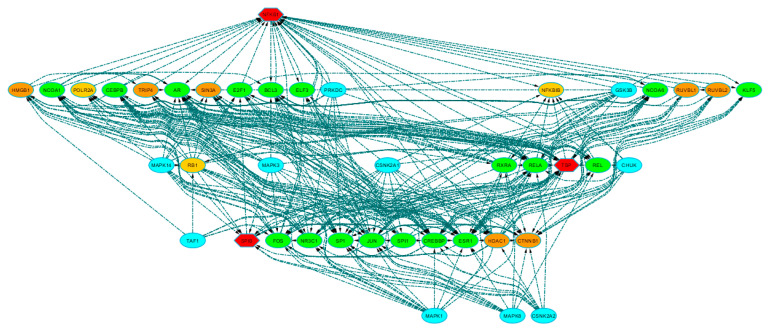
Extended regulatory network. The network shows the relationships (directed dashed edges) among the hub transcriptional factors, the additional proteins inferred by protein–protein interactions, and the most significant kinases identified by kinase enrichment analysis. Node shapes distinguish hub TFs (hexagon) from all the other inferred proteins (ovals). Node colors reflect node types or functions: hub TFs = red; inferred TFs = green; kinases = light blue; TcoFs = orange; inferred molecules with other functions = light orange. Also see Appendix A for detailed annotation.

**Figure 4 biomedicines-08-00288-f004:**
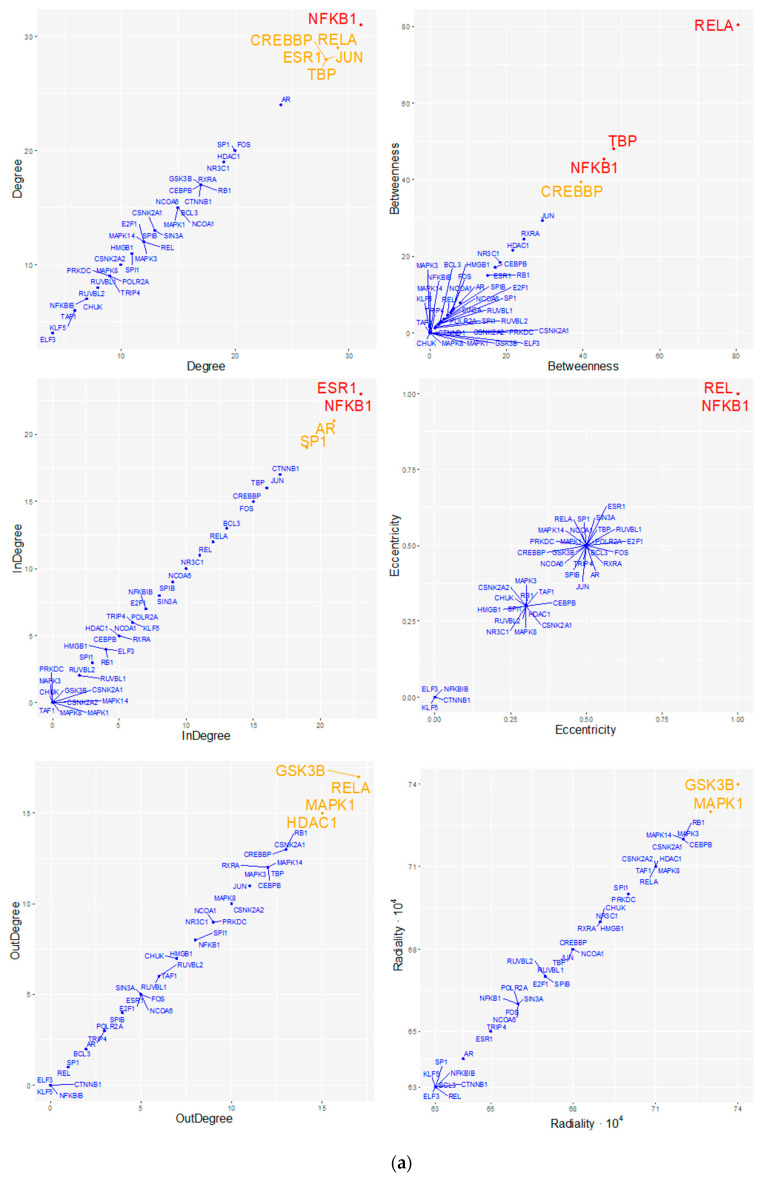
(**a**) Topological analysis of the extended regulatory network. Each panel represents the scatterplot of the index-over-index topological measures calculated for the extended regulatory network. To draw the representative scatterplots, only one index was chosen among those that were highly correlated and with very similar meanings. The presence of a non-uniform distribution of nodes, with most nodes having a low index and a few having a high index, identifies those nodes that clearly differ from the average measure of the index throughout the network (red and orange dots or labels). Red, orange, and blue colors, respectively, refer to nodes with z-scores > 2.0, between 1.5 and 2.0, and <1.5. Detailed results are reported in Appendix A. (**b**) Correlation plot of the topological indexes. Indexes with similar meanings show a highly significant Pearson’s correlation (*p*-values < 0.001). Numbers and circle sizes (from smaller to bigger) refer to correlation coefficient *r* values. Positive and negative correlations are displayed by a gradient color from white (low) to blue (high) and from white (low) to red (high), respectively. To simplify visualization, crosses mark non-significant correlations.

**Figure 5 biomedicines-08-00288-f005:**
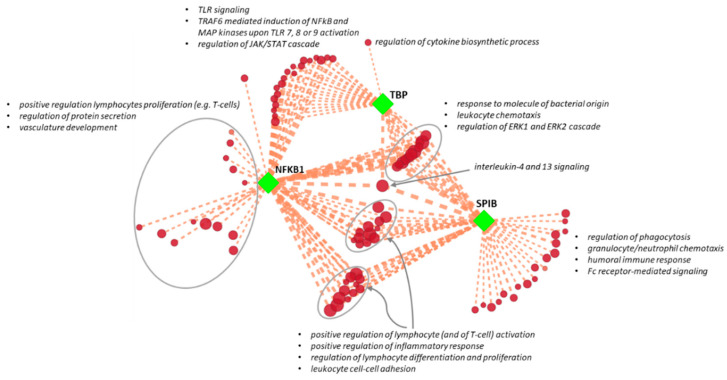
Enrichment network of transcriptional clusters. NFKB1, SPIB, and TBP transcriptional clusters (squared green nodes) and pathogenetic Gene Ontology–biological process (GO-BP) pathways associated with abdominal aortic aneurysm perivascular adipose tissue (red nodes) are connected by dashed orange edges, which indicate significant overlaps between two gene sets. Edge thickness (from thinner to thicker) is proportional to the number of overlapping genes. GO-BP pathway node colors (from lighter to darker) and sizes (from smallest to bigger) are proportional to the original normalized enrichment score calculated by gene set enrichment analysis. Groups of redundant gene sets were manually circled and labeled by relevant overview GO-BP pathways terms.

**Table 1 biomedicines-08-00288-t001:** Summary of enriched motifs and tracks aggregated in clusters.

Cluster	TF	NES	AUC	# Targets	# Motifs/Tracks
M1	TBP	6.20	0.098	74	5
M2	NFKB1	5.91	0.095	91	7
T1	CHD1	5.23	0.130	29	1
M3	SPIB	5.12	0.085	104	4
M4	SRF	4.66	0.080	53	6
M5	BCL6	4.14	0.074	55	3
M6	PAX3	4.07	0.073	48	2
M7	ATF2	3.58	0.068	46	1
M8	CEBPE	3.53	0.067	50	2

Transcription factors (TFs) with direct annotation, normalized enrichment score (NES), and area under the cumulative recovery curve (AUC) values refer to the highest enriched motif or track of each cluster. Motif (M) and track (T) clusters are ordered by NES. #, number of.

**Table 2 biomedicines-08-00288-t002:** Summary of topological indexes used to evaluate the centrality of each candidate TF in the reconstructed gene regulatory network.

TF (Cluster)	Degree	Betweenness Centrality	Radiality
SPIB (M3)	104	0.36	3.41
NFKB1 (M2)	91	0.22	3.24
TBP (M1)	74	0.14	3.02
BCL6 (M5)	55	0.08	2.77
SRF (M4)	53	0.08	2.75
CEBPE (M8)	50	0.08	2.71
PAX3 (M6)	48	0.05	2.68
ATF2 (M7)	46	0.06	2.65
CHD1 (T1)	29	0.04	2.43

Candidate transcription factors (TFs) of each cluster are ranked by degree. The selected “hub” TFs are highlighted with bold characters.

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
