# Peer review of "Gene Regulatory Network Analysis of Perivascular Adipose Tissue of Abdominal Aortic Aneurysm Identifies Master Regulators of Key Pathogenetic Pathways"

_biomedicines, 2020, doi:10.3390/biomedicines8080288_

Round 1

Reviewer 1 Report

The manuscript by Piacentini et al entitled "Gene regulatory network analysis on perivascular adipose tissue of abdominal aortic aneurysm identifies master regulators of key pathogenetic pathways" elegantly identified 172 unique genes in the PVAD of human AAA. The authors outlined putative targets for AAA interfering treatment options for AAA.

Some minor comments are suggested to enhance the clinical significance of the study:

  1. How does heterogeneity of the PVAD around the different areas of the aortic tissue may influence the analysis
  2. Because of activation of different pathways at different stages of AAA pathogenesis, how these findings will be therapeutically effective.
  3. How does communication between adipose tissue and inflammatory cells affect the expression of TFs/downstream genes.

Author Response

We are grateful for the constructive suggestions that originated from the Editors and the Reviewers revision and we hope to have responded satisfactorily to their comments.

Response to Reviewer 1

We thank the Reviewer 1 for his/her precise and helpful observations.

Answers to specific questions:

  1. How does heterogeneity of the PVAD around the different areas of the aortic tissue may influence the analysis

We thank the reviewer for this observation as it allows us to specify an important aspect of our study design. Although we agree with the concept that a different tissue sampling might produce some variations in gene expression data, we are confident that our results are robust and little affected by tissue heterogeneity for the following reasons. First, gene expression data were obtained from RNA extracted from 50-100 mg of adipose tissue (for each sample) to ensure that the specimen was representative enough of the adipose tissue “state” in the specified area. Second, in our work on aortic occlusive disease (AOD) (see Piacentini et al. 2020, doi: 10.1038/s41598-020-63361-5), we compared PVAT in the same tract of the abdominal aorta, i.e. the adipose layer surrounding the distal aorta (atherosclerotic lesion) with the proximal aorta (plaque-free segment) but we did not detect any locally restricted gene-expression patterns. This would mean that the variations found in the AAA were, however, reliable and descriptive of the disease. Third, the study on AAA is based on a paired sample design on biological replicates, which overall should provide better control of the confounding effects of interindividual variation and of tissue sampling, thus, improving both the overall sensitivity and statistical power of the analysis (see the original article by Piacentini et al., 2019: doi: 10.1161/ATVBAHA.118.311803).

In summary, we would expect very similar results even from different samples in different areas of the diseased vs the healthy tissues.

  1. Because of activation of different pathways at different stages of AAA pathogenesis, how these findings will be therapeutically effective.

We thank the Reviewer for this observation. In the present manuscript, the identified master regulators and putative therapeutic targets are mainly associated with an altered adaptive immune response, specifically lymphocyte activation/proliferation. In our previous work, while showing that the activation of other pathways may concur to the disease evolution, we assumed that immune response processes were critical to sustaining a chronic inflammatory loop that characterizes AAA, since they were active in both the early and later stages of AAA (cf. Figure 6 by Piacentini et al. 2019: doi: 10.1161/ATVBAHA.118.311803). Thus, it is tempting to speculate that interfering with those specific pathogenetic pathways could have an overall beneficial effect, as it could limit a key detrimental process that seems to be essential in all the phases of AAA.

We integrated the discussion section outlining this important point suggested by the Reviewer (lines 385-391 of the revised manuscript).

  1. How does communication between adipose tissue and inflammatory cells affect the expression of TFs/downstream genes.

We thank the Reviewer also for this suggestion. Briefly, adipose tissue and immune cell infiltrates can communicate in several ways by influencing each other through the activation of different signaling pathways. For example, adipocytes can secrete soluble factors, e.g. adipokines, chemokine or inflammatory interleukins that may activate signaling mechanisms on target cells that require, for instance, the activation of the NF-kB complex. Immune cells, in turn, may act through the production of inflammatory molecules and/or by recognizing antigens presented by the HLA class I or II expressed on adipocytes. Finally, the relationship between adipocytes and immune cell infiltrates may be influenced by their interaction with the cellular and structural component of the vessel because the perivascular adipose tissue is an anatomical continuum that makes direct contact with the adventitia of the vessel (see Kim et al.,2009: doi: 10.1161/ATVBAHA.119.312304; Rajsheker et al. 2010: doi: 10.1016/j.coph.2009.11.005; Huh et al. 2014: doi: 10.14348/molcells.2014.0074).

We included a comment concerning this point in the discussion section (lines 376-382 of the revised manuscript).

Reviewer 2 Report

The aim of the study is to gain understanding of the etiology of AAA. For that purpose the authors choose to study the expression profiles of LVAT.
In their previous studies they compared expression profiles of dilated to non dilated aorta LVAT tissue and stenotic from arteriosclerotic tisuues.
To understand the content of the current study, the previous study ref 6 needs to be read . Unfortunately the supplementary information of that paper could not be downloaded.
The current study focusses on the differentially upregulated genes in the LVAT of dilated aorta. And doing so uses a hypotheses free approach on aneurysm. The authors fail to address the existing knowledge on the genetics of AAA, and at least the 40 actionable aneurysm genes that have been identified sofar and are used in clinical settings for genetic diagnostics.
Nether the previous study or this one, reflects on the discrepancy between their gene selection and the well-known genetic causes, involving ao extracellular matrix remodeling, imbalance of mechano transducting signaling and.
Another issue is that LVAT was used from 30 patients, in a period where the majority of patient had EVAR procedures and only in rare complicated cases an open procedure was performed. Raising the question whether there was a selection of atypical AAA cases used for the study. Which may have influenced the results.
AAA is genetically and etiologically heterogeneous, and the genotypes of the study population should be included.
In conclusions the authors need to put their findings in perspective of the genetic etiologies for AAA.

Author Response

We are grateful for the constructive suggestions that originated from the Editors and the Reviewers revision and we hope to have responded satisfactorily to their comments.

Response to Reviewer 2

We thank the Reviewer 2 for his/her observations.

Answers to specific questions:

  • To understand the content of the current study, the previous study ref 6 needs to be read . Unfortunately the supplementary information of that paper could not be downloaded.

We regret this technical issue. We do not know the reason why the Reviewer was not able to download the Supplementary information of our previous work because it is freely available at https://www.ahajournals.org/doi/suppl/10.1161/ATVBAHA.118.311803.

  • The current study focusses on the differentially upregulated genes in the LVAT of dilated aorta. And doing so uses a hypotheses free approach on aneurysm. The authors fail to address the existing knowledge on the genetics of AAA, and at least the 40 actionable aneurysm genes that have been identified sofar and are used in clinical settings for genetic diagnostics.

The authors understand the importance of genetics in aneurysmal diseases and we are aware that, in thoracic aortic aneurysms, genotyping can be used to identify at-risk individuals and guide disease management. Conversely, to the best of our knowledge, the genetic associations that have been identified for abdominal aortic aneurysms cannot yet be used for clinical purposes. Consistently, also the international guidelines on the diagnosis and treatment of aortic diseases do not report any clinically validated genetic diagnostics for abdominal aortic aneurysms (please see Pinard et al. 2019: doi: 10.1161/CIRCRESAHA.118.312436; and European Heart Journal (2014) 35, 2873–2926: doi:10.1093/eurheartj/ehu281). We believe that the Reviewer refers to other syndromes affecting the aorta but that are not the object of this study.

  • Nether the previous study or this one, reflects on the discrepancy between their gene selection and the well-known genetic causes, involving ao extracellular matrix remodeling, imbalance of mechano transducting signaling and.

We are sorry to disagree with the Reviewer for this specific point. In our opinion, there is not any clear discrepancy between the differentially expressed genes that we have observed in our works and “well-known genetic causes” of AAA. This is because to date, in AAA, only associations that involve quite a few genes have been reported and none of them can be assumed as causative for AAA (see Bradley et al. 2015: doi: 10.1016/j.ejvs.2015.09.006; and Jones et al. 2017: doi: 10.1161/CIRCRESAHA.116.308765). However, we wish to highlight that, in our previous work, we also found differentially expressed genes related to extracellular matrix remodeling, accordingly with the Reviewer observation that pointed to these genes as key players in the AAA context.

  • Another issue is that LVAT was used from 30 patients, in a period where the majority of patient had EVAR procedures and only in rare complicated cases an open procedure was performed. Raising the question whether there was a selection of atypical AAA cases used for the study. Which may have influenced the results.

We thank the Reviewer for this observation since it allows the authors to clarify this point. AAA patients included in the study underwent elective surgery between 2010 and 2014 at the Centro Cardiologico Monzino IRCCS, Milan. Elective repair of AAA was performed in compliance with the international and national guidelines for the treatment of AAA patients (Moll et al. 2011, doi: 10.1016/j.ejvs.2010.09.011; Pratesi et al. 2016, https://www.minervamedica.it/en/journals/vascular-endovascular-surgery/issue.php?cod=R46Y2016S01) and, thus, cannot be considered as atypical AAA cases.

We specified this point in the “Materials and Methods” section (lines 416-419 of the revised manuscript).

  • AAA is genetically and etiologically heterogeneous, and the genotypes of the study population should be included.
    In conclusions the authors need to put their findings in perspective of the genetic etiologies for AAA.

We agree with the Reviewer that the genetic aspect of AAA can be important. We added comment concerning this point in the discussion section (lines 408-411 of the revised manuscript).